# The impact of long-term care insurance on family care for older adults: The mediating role of intergenerational financial support

**Lianjie Wang**[1], **Jing Liu**[2]*

**1** Department of Sociology, Jiangnan University, Wuxi, China, **2** School of Public Administration, Zhejiang University of Finance and Economics, Hangzhou, China

* jliu65@zufe.edu.cn (JL)

**Data Availability Statement:** Data used in this paper is third party data, which is publicly available for all upon request at http://charls.pku.edu.cn/.

**Funding:** This research is supported by the National Social Science Fund Project in China

## Abstract

Rapid population aging has been placing heavy tolls on Chinese family caregivers. Previous empirical evidence from multiple countries have shown that establishing national long-term care insurance was effective in reducing family care burdens. Utilizing data from the China Health and Retirement Longitudinal Study (CHARLS) wave 2011 to 2018, this study examined the effects of implementing the pilot long-term care insurance program on family care received by the Chinese older adults, by using a time-varying Difference-in-Differences (DID) method. The results showed that: (1) the implementation of the pilot long-term care insurance program has led to a 17.2% decline in general for family care received by the Chinese older adults. (2) The effect of participating in the pilot program on family care received differed by respondent's household registration, health status, marital status, and possesion of retirement pension, and were specifically pronounced among those who were urban residents, having spouse, living with disabilities, and living with no retirement pension. (3) Further results from the mechanism analyses showed that the pilot long-term care insurance program decreased the level of family care by reducing the dual intergenerational financial support between older adults and their adult children. (4) Although participating in the pilot program decreased older adult's dependence on their adult children, their physical and mental health status were not negatively affected. This study contributes to the existing literature by evaluating the effects of implementing the pilot long-term care insurance program on family care received by the Chinese older adults, and lends supports to the previous studies that participating in long-term care insurance significantly reduces old adults' demand for family care, but not in sacrifice of their physical and mental well-being.

## Introduction

China has undergone severe population aging since 2000. According to the National Bureau of Statistics of China, older adults aged 60 and above has increased from 194 million in 2012 to 280 million in 2022, with an average annual growth of approximately 7.8 million. Previous research projected that China's older population will surpass 400 million by the mid 21

(21BSH163). The funders had no role in study design, data collection and analysis, decision to publish, or preparation of the manuscript.

**Competing interests:** The authors have declared that no competing interests exist.

century, representing 34% of its entire population [1]. Over the past decade, the elderly population aged 80 and above in China even experienced a faster annual growth rate (4.7%) as compared to their counterparts who were aged 60 and above [2]. Older adult's physical function is gradually declining due to such trend of aging population, resulting in a rapid increase in the size of the disabled and partially disabled elderly population. Under such context, the increasing prevalences of chronic diseases and disabilities have become two distinctive characteristics in current China. According to the "Blue Book on Aging: China's Livable Environment for the Elderly Development Report," the number of disabled older adults in China surpassed 40 million by the end of 2015. This marked an increase of approximately seven million from 2010, representing 19.5% of China's total elderly population. According to data from China's latest population census, there are approximately 267 million older adults in China. Among them, approximately 180 million are currently experiencing chronic diseases and more than 60 million are either fully disabled or partially disabled (National Bureau of Statistics, 2020). More data show that, among older adults with disabilities, 52% faced challenges in self-care and 88% experienced limitations in activities of daily living [3]. The implementation of the pilot long-term care insurance (LTCI) program were to cope with these challenges brought by China's population aging and the escalating need for long-term care (LTC) services.

China's old age care provision for a long period of time adheres to a priority sequence pattern of "spouse-children-relatives-society". Informal care provided by family members primarily constitutes the prior and primary form of the long-term care for older adults [4]. However, significant time, economic, and opportunity costs associated with leaving work for family care have become increasingly unaffordable for modern Chinese families. Moreover, the overlapping of generational relationships and the absence of specialized nursing have further diminished the efficacy of family care. Studies have indicated that engaging in family care activities for older adults is associated with decreasing labor participation rate, particularly for female caregivers. This reduction in labor participation rate was estimated to be approximately 12.46%, resulting in a decrease in labor income by approximately 7.21% for informal caregivers [5]. Due to the rising number of older adults with disabilities, the escalating costs of caregiving have become unaffordable for most families. Establishing a comprehensive long-term care system is essential to meet the professional nursing needs of older adults, which in returen alleviates the burden of family care, and improves the labor participation rate for Chinese families.

From a global perspective, in response to the challenges posed by aging population, countries such as Germany and Japan have implemented LTCI systems that significantly impacted the daily lives of their disabled older adults. In 2016, the Chinese government officially initiated the pilot implementation of a national LTCI system by issuing the Guiding Opinions. This initiative was launched in 15 cities across the country. In 2020, the National Medical Insurance Administration expanded the number of pilot cities to 49. The 20th National Congress Report in 2022 reiterated the importance of establishing a LTCI system. The development of China's LTCI system is presented through: (1) the evolutionary developing stages of "partial pilot, expanding pilot, and an increasing awareness of establishing a national system". (2) transitioning from the initial pilot stage 1.0 to the comprehensive national promotion stage 2.0. LTCI not only offers financial support for individuals with physical or cognitive impairments that hinder their ability to carry out daily activities, but also covers services for both institutional and family care. Currently, China's LTCI pilots have been in operation for five years. Evaluating the policy impact of the pilot program and assessing its effectiveness in alleviating the burden of family care hold significant practical importance.

This study particularly focuses on the pilot cities of China's national LTCI program from 2012 to 2018 and evaluates the influence of the pilot LTCI program on family care received by

the Chinese older adults. This study makes three significant contributions. Firstly, due to the pilot nature of the LTCI program and limited data availability, relevant research about the impact of the LTCI program on family care for older adults have been lacking. This study explores the average treatment effect, heterogeneity effects, meanwhile expands the existing research scope by analyzing the influencing mechanism of China's pilot LTCI program on family care received by the Chinese older adults. This study also expands the existing literature by examining whether the change of family care level negatively affects older adults' health status. Secondly, the data of this study come from China's national representative dataset CHARLS, which was collected from a national quasi-natural experiment by implementing the pilot LTCI program. This study applies the time-varying difference-in-differences (DID) model, the propensity score matching-DID (PSM-DID) method, and various robustness test methods to meticulously examine the potential causal relationship between the pilot LTCI program and family care received by the Chinese older adults. Findings from this study can serve as a valuable reference for evaluating the policy effectiveness of the LTCI pilot program. Thirdly, most relevant studies focused on exploring the impact mechanism of LTCI on family care through intergenerational support within families. However, there is a need for additional research to examine the policy effects of the LTCI pilots. This paper examines the policy effectiveness of the pilot LTCI program from the perspectives of intergenerational support and health outcomes. It serves as a valuable reference for the strategic planning of social policies regarding a future comprehensive implementation of the national LTCI in China.

## Policy background and research hypotheses

### Policy background

Before the pilot LTCI program in China, over 90% of the Chinese older adults relied on family care. When older adults encountered illnesses, medical care provided by medical insurance was typically used for treating short-term illnesses. When older adults faced the risk of disability, LTC services were usually provided by spouses, children, or other relatives. Although both the LTCI and medical insurance fall under the category of health insurance and aim to provide financial support and service coverage to address the high costs associated with health issues in China, there are significant differences between the two. Firstly, in terms of coverage, the LTCI primarily concentrates on providing coverage for daily living assistance and long-term care, specifically tailored for individuals with fuctional limitation and ongoing care needs. In contrast, medical insurance primarily aims to cover the costs associated with sudden or acute illnesses. Secondly, in terms of payment systems, the LTCI primarily adopts a typical LTCI format, which allows individuals to choose different insurance terms based on their insurance plans and personal needs. On the other hand, medical insurance primarily takes a short-term insurance format, which can be either reimbursement-based or fixed-sum payment-based. Overall, public medical services for the LTC in China suffer from significant deficiencies, low reimbursement rates for basic medical insurance in particular. Consequently, older adults shoulder a greater share of medical expenses, resulting in a notable prevalence of "poverty due to illness" and the recurrence of poverty due to subsequent illnesses among this population. Therefore, establishing a comprehensive LTCI system becomes imperative in meeting the LTC needs for older adults and alleviating Chinese families' care burdens.

Policies related to the pilot LTCI program in China can be roughly divided into three stages: the embryonic stage (2006–2012), the initial growth stage (2012–2015), and the comprehensive development stage (after 2016). In the embryonic stage of the pilot program, the central and local governments in China introduced various policies pertaining to the provision of LTC. These policies encompassed a range of initiatives, such as the provision of subsidies for older

adults, the establishment of specialized facilities for elderly care, and the promotion of end-of-life care services. In 2006, the National Committee on Aging issued the "Opinions on Accelerating the Development of the Elderly Service Industry," which set forth a comprehensive framework for supporting and advancing the growth of elderly care and retirement service sectors. During the period of 2006–2007, the offices responsible for aging across various provinces and cities sequentially implemented the "Eleventh Five-Year Plan for the development of the Elderly Care Industry." Unfortunately, the plan did not lead to a comprehensive recognition and awareness of treating LTC as an independent institutional framework. Nevertheless, implementations of the relevant policies aiming at enhancing elderly care services have played a crucial role in establishing a stable care system for disabled individuals. These efforts have provided a solid groundwork for the subsequent inception of China's LTCI system.

During its initial growth stage phase, the need of developing the LTCI began to gain attention and recognition from the Chinese government. Within four years, numerous policies were introduced with the aim of promoting a comprehensive social LTCI system. For instance, in 2013, the government made revisions to the "Law of the People's Republic of China on the Protection of the Rights and Interests of the Elderly", incorporating specific service provisions pertaining to the LTCI. Furthermore, in 2015, the General Office of the State Council issued the "National Healthcare Service System Planning Outline 2015–2020", which underscored the importance of supporting the development of the LTCI. These policy efforts played a pivotal role in laying the foundation for the subsequent implementation of a stable LTCI system. In addition to the national-level policies, the cities of Qingdao, Shanghai, Nantong, and Changchun have also introduced policies to facilitate the establishment of a social LTCI system.

Since 2016, China's LTCI has entered a period of comprehensive development. In 2016, the Ministry of Human Resources and Social Security issued the "Guiding Opinions on Pilot Implementation of the LTCI", selecting 15 cities, including Beijing, as the first batch of international-level pilot cities. In 2020, the National Healthcare Security Administration and the Ministry of Finance issued the "Guiding Opinions on Expanding the Pilot Implementation of the LTCISystem," determining the second batch of national pilot areas. The number of pilot cities and regions for the LTCI pilots further reached to 49 at that time point. In 2022, the 20th National Congress officially proposed the establishment of a LTCI system.

As of March 2023, a total of 62 cities in China have implemented LTCI pilot program [6]. The government has strategically designated pilot cities in the eastern, central, and western regions to ensure regional coverage balance and maximize the effectiveness of the pilot experience. However, the pilot program is influenced by various factors such as regional experience, developmental level, availability of medical resources, government governance efficiency, and the degree of population aging. This paper utilizes data from a national representative dataset CHALRS, specifically the wave 2011 to 2018 to assess the impact of the pilot LTCI program on family care received by the Chinese older adults. Fig 1 illustrates the cities that participated in the pilot LTCI program in China from 2012 to 2018.

## Literature review and research hypothesis

There are three perspectives regarding the relationship between LTCI and informal family care. The first argument suggests that LTCI reduces the need for family care. In the cases of United States and Japan, with access to formal care services, older adults may receive less informal support from their families, and formal care can effectively alleviate the burden of family caregivers [7,8]. Pauly argued that the moral motivation within families sets informal caregivers apart from traditional moral hazards, making them the primary source of care for older adults [9]. Older adults who purchase LTCI can receive financial assistance to cover a portion

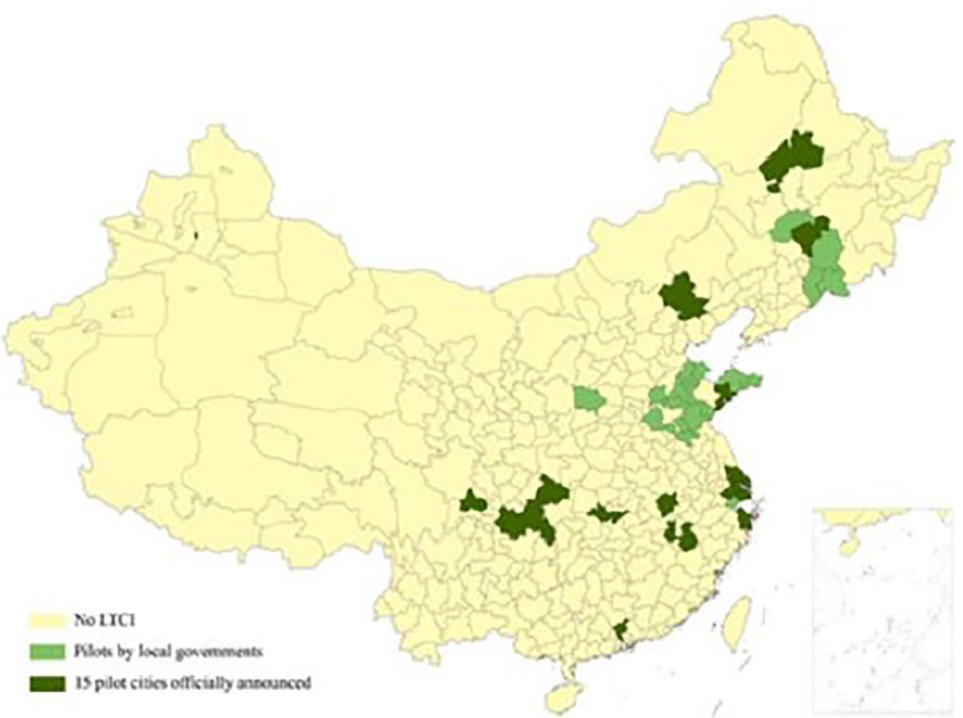

**Fig 1. Pilot cities for the pilot LTCI program in China (2012~2018).** *Notes.* The chronological order of the pilot cities for the pilot LTCI is as follows: 2012: Qingdao. 2013: Shanghai. 2014: Dongying. 2015: Rizhao, Changchun. 2016: Jinan, Shangrao, Chengde, Songyuan, Jilin, Nntong, Jingmen. 2017: Anqing, Xuzhou, Chengdu, Guangzhou, Linyi, Liaocheng, Tai'an, Linfen, Qiquhar, Chongqing, Ningbo, Mehekou, Tonghua, Baishan, Shihezi, Jiaxing, Suzhou. 2018: Binzhou, Zibo, Heze, Zaozhuang, Yantai, Weihai. The timing and coverage of the pilot cities were determined based on the ploicies published on the government websites of each city.

of the formal care expenses. This insurance option reduces the financial burden of older adults and releases their family members from providing informal care as they previously did. The second perspective argues that LTCI improves family care. In England, when older adults receive formal and specialized LTC services, their family members can also provide informal care for them, such as daily life assistance, due to the presence of altruistic motivations and family norms and moral awareness [10,11]. Strong adherence to family norms encourages family members to provide informal care, thus better addresses older adult's LTC needs [12]. Thirdly, a study in Europe showed that LTCI and family care complement each other and have distinct roles in the aging care process [13]. Formal care specializes in providing specialized services, while family care focuses on general daily life care services. However, it is important to note that the relationship between the two is dynamic [14]. This paper argues that scholars should not rely solely on theoretical interpretation and must thoroughly examine realities. In China, the reality is that the number of disabled older adults is increasing, and family care is the primary LTC option for most older adults. Under such context, we propose the following hypothesis:

Hypothesis 1: LTCI decreases older adult's informal care received from families in China.

The existing literature have extensively studied the heterogeneity effect of the pilot LTCI program on family care received by the Chinese older adults, which can be categorized into three aspects of factors: demographic factors, family factors, and social factors. The utilization of formal and informal care varies significantly based on individual factors such as marital

status [15], level of disability [16], and household registration type [17]. The impact of LTCI on family care utilization may be more pronounced for older adults who are married, have severe disabilities, and reside in urban areas. Among family factors, the impact of LTCI on family care varies significantly based on living arrangements [18] and types of caregivers [19]. Among social factors, the impact of LTCI on family care is also influenced by institutional compensation. According to a European study conducted by Courbage et al., LTCI has been found to decrease older adult's reliance on family care in Spain, but increase the reliance in Italy [12]. This finding highlighted the varying impact of LTCI on family care across different countries. This discrepancy can be attributed to the difference in the LTCI models employed by the two countries: Spain utilizes the reimbursement model, whereas Italy follows the cash subsidy model. This paper presents an argument for the importance of considering heterogeneity when analyzing the impact of LTCI on family care received. In light of this argument, the paper puts forth the following research hypothesis:

Hypothesis 2: Heterogeneity effects exist in the impact of LTCI on family care received.

The intergenerational reciprocity theory posits that family members have a responsibility for resource exchange and mutual support, which operates through a "feedback mode" of two-way communication and balanced reciprocity. On one hand, the provision of family care for children by parents through financial support or inheritance can be understood as a "contractual relationship" between generations, in which parents provide support for children at an early stage in exchange of upward support and care in old age [20]. LTCI, as a part of the social security system, offers benefits such as reducing the risk of disability for older adults, improving their level of old age security, and reducing their dependence on informal care from children and grandchildren. The provision of downward financial support from older adults to the next generation may serve as a crucial mechanism through which LTCI influences family care. LTCI diminishes the incentive for intergenerational exchange among older adults, leading to a reduction in financial support for the next generation and consequently impacting their receiving of informal care from adult children [21]. In China, providing care for older adults is influenced by family collectivism and self-sacrifice motives. Altruistic motivation has been the main factor that can optimize the allocation of resources within the family, thus maximizing the family's interests over personal interests [22]. On the other hand, adult children frequently engage in a trade-off between "providing care" and "providing financial support" when fulfilling the demands for care for older adults within the family. If adult children have access to a higher level of income from the labor market, they may be prone to provide financial support instead of physical informal care. However, if family members are unable to earn enough income from the labor market for upward financial support, they may reduce their labor participation and fulfil their caregiving responsibilities for older parents [23]. Especially when the traditional concept of family support is dimishing, there is a growing trend among adult children to reduce their physical caregiving responsibilities by offering financial assistance. In other words, the financial support provided by adult children to older adults may serve as a mechanism through which LTCI influences family care. LTCI can potentially incentivize children to offer financial assistance, thereby compensating for their absence of their physical caregiving. Building upon this logic, the present study puts forth the following research hypothesis:

Hypothesis 3: LTCI affects family care provision through "downward" (older adults to children) financial support

Hypothesis 4: LTCI affects family care provision through "upward" (children to older adults) financial support

The contribution of this study to the existing literature can be summarized as follows: Firstly, unlike the previous studies that have solely examined the theoretical impact of LTCI on

family care provision, this study provides empirical evidence to test the theories. Drawing on the quasi-natural experimental data from China's LTCI pilot program, this paper employs the time varying Difference-in-Differences (DID) method to empirically examine the influence of LTCI on family care received in China. By doing so, this study addresses the limitations of previous empirical research in this area. Secondly, the existing literature needs a thorough analysis of the impact pathway through which LTCI affects family care provision, particularly concerning the influence of formal care on intergenerational relations within families. This study investigates the mechanism of influence from the perspective of reciprocal intergenerational financial support between older adults and children. This analysis aims to contribute valuable insights for integrating formal and informal care. Third, this paper further discusses the crowding-out effect of LTCI on family care. If the LTCI reduces family care, does that mean that there is a decline in the overall quality of care for older adults? This paper also adds to the existing literature by analyzing the impact of the LTCI on older adults' physical and mental health.

## Materials and methods

### Data

The data utilized in this paper come from the China Health and Retirement Longitudinal Study (CHARLS), which was conducted between 2011 and 2018. This project was hosted by the National School of Development at Peking University and was jointly implemented by the China Social Science Survey Center of Peking University and the Youth League Committee of Peking University. The baseline data was collected in 2011, which was then followed by subsequent follow-up surveys in 2013, 2015, and 2018. By 2018, researchers had successfully surveyed nearly 20,000 respondents residing in 450 communities across 28 provinces, autonomous regions, and municipalities directly under the Central Government (Excluding Hong Kong, Macau and Taiwan Province). The questionnaire design drew on multiple international survey design experiences, including the Health and Retirement Study (HRS) in the United States, the English Longitudinal Study of Aging (ELSA), and the Survey of Health, Aging, and Retirement in Europe (SHARE). The project employed a multi-stage sampling approach, with the use of probability proportional to size (PPS) sampling at both the county/district and village levels. CHARLS pioneered the use of electronic mapping software (CHARLS-GIS) to create village-level sampling frames using a mapping method. We chose CHARLS for the following reasons. Firstly, CHARLS effectively tracked survey respondents with a high response rate, comprehensive sample coverage, and a large-scale of sample size. The quality of the data collected was widely valued by the international academic communities. Secondly, CHARLS data provided relevant variables for this study, including basic personal information, family structure, economic support, family care, health status, and long-term care participation status. In this paper, the data were processed as follows: (1) We focused on individuals aged 60 and above as the research subjects, and 18,816 respondents were included. (2) The CHARLS database covers four waves of data from 2011 to 2018. Table 1 shows the implementation time of policies in various pilot areas from 2012 to 2018. Policy changes after 2018 are not considered in this study. After processing the data, 2,094 respondents were deleted. This study obtained a total of 18,948 respondents, among which there were 1,291 respondents in the treatment group and 17,657 in the control group.

### Meaurement

**The dependent variable.** The dependent variable in this study is family care received, quantified as the "average number of hours receiving care from family members in the past

**Table 1. Descriptive statistics.**

| Variables | All samples | | Treatment group | | Control group | | Before intervention | | After intervention | |
|---|---|---|---|---|---|---|---|---|---|---|
| | Mean | SD | Mean | SD | Mean | SD | Mean | SD | Mean | SD |
| Family care (hours/year) | 649.873 | 249.136 | 547.107 | 185.084 | 657.387 | 269.177 | 518.208 | 156.279 | 686.903 | 319.882 |
| Treatment variable (treatment group = 1) | 0.068 | 0.247 | 1 | 0 | 0 | 0 | —— | —— | —— | —— |
| Age (years) | 69.958 | 6.767 | 72.108 | 6.488 | 69.796 | 6.760 | 67.775 | 6.514 | 72.411 | 6.465 |
| Gender (Male = 1) | 0.480 | 0.500 | 0.480 | 0.499 | 0.480 | 0.500 | 0.480 | 0.500 | 0.481 | 0.500 |
| Household registration (rural = 1) | 0.603 | 0.489 | 0.727 | 0.446 | 0.594 | 0.491 | 0.735 | 0.526 | 0.773 | 0.419 |
| Marriage status (with spouse = 1) | 0.751 | 0.433 | 0.740 | 0.439 | 0.752 | 0.432 | 0.780 | 0.414 | 0.713 | 0.452 |
| Education level (years) | 3.601 | 3.237 | 4.014 | 3.836 | 3.571 | 3.185 | 3.941 | 3.836 | 3.911 | 3.798 |
| Health status (Health = 1) | 0.877 | 0.328 | 0.849 | 0.358 | 0.879 | 0.326 | 0.913 | 0.282 | 0.844 | 0.363 |
| Endowment insurance (have = 1) | 0.780 | 0.414 | 0.812 | 0.391 | 0.778 | 0.416 | 0.793 | 0.405 | 0.849 | 0.358 |
| "Upward" economic support (Yuan/year) | 2840.333 | 10648.831 | 3514.305 | 13613.512 | 2791.055 | 10437.293 | 2083.246 | 9551.388 | 3154.163 | 9238.366 |
| "Downward" economic support (Yuan/year) | 1208.358 | 11439.480 | 1875.772 | 11731.260 | 1159.559 | 11416.671 | 810.396 | 8153.859 | 1369.925 | 15545.83 |
| N | 18948 | | 1291 | | 17657 | | 14210 | | 4738 | |

*Notes.* Three decimal places were retained after rounding off decimal numbers. In the subsequent empirical study, logarithms were used for family care received, "upward" financial support, and "upward" financial support. We used the interpolation method to handle missing data by predicting them based on neighboring observations.

year" as assessed by the CHARLS questionnaire. This study computed the total number of care hours received from family members, including parents, spouses, and children. As family care was treated as a continuous variable, a logarithmic transformation was conducted in the statistical analysis.

**The independent variable.** The independent variable in this study was the LTCI participation. If a region implements the LTCI pilot program, the value is assigned as 1; otherwise, it is assigned as 0. Due to the non-uniform implementation time and inconsistent coverage subjects in each pilot area, we adjusted the treatment group and control group based on changes in pilot time and coverage for each city as shown in Fig 1. For instance, in the case of urban samples from Qingdao, the control group was represented by the year 2011, while the treatment group included data from 2013 onwards. In contrast, the rural samples initially served as the control group prior to 2015, but were later included in the treatment group from 2015 onwards. Similarly, the treatment and control groups were defined for other regions based on their respective pilot times and coverage.

**The control variables.** Based on the study conducted by Lei et al. [24], this paper begins by examining the micro-level factors influencing family care. Age, gender, household registration, education level, marital status, health status, and endowment insurance were chosen as the control variables for this study. The age variable was calculated as the difference between the survey year and the year of birth of the respondents. Gender was coded as 1 for males and 0 for females. Household registration was categorized as 1 for urban and 0 for rural. Education level represented the number of years of education. Marital status was coded as 1 for individuals with a spouse and 0 for individuals without a spouse. Health status was coded as 1 for individuals in good health and 0 for individuals with disabilities. Retirement pension was coded as 1 for individuals who have it and 0 for individuals who do not.

**The mechanism variables.** The study includes two mechanism variables. The first is referred to as "upward" financial support, measures the financial assistance provided by adult children to older adults. This variable is assessed by survey questions regarding the financial

support received from children who live in different locations within the past year. This paper calculates the total amount of financial support by aggregating the contributions from all children. Secondly, the term "downward" financial support refers to the financial assistance provided by older adults to their children. It is measured through the "financial support given to children residing in different locations during the past year" as reported in the questionnaire. Additionally, this paper aggregates all the collected data to derive the total amount. The mechanism variable is a continuous variable, measured in yuan per year in the statistical analysis. The logarithm transformation was conducted for better data interpretation.

## Methods

**Time-varying DID model.** The pilot LTCI program implemented in various provinces and cities in China offer a quasi-natural experimental setting for this study. Due to variations in individuals, policies, and time across different city pilots, accurately assessing the impact of the LTCI on elderly family care requires the use of a time-varying Difference-in-Differences (DID) model to determine the policy effect. The Time-varying Difference-in-Differences (DID) model possesses strong applicability and effectively addresses endogeneity issues arising from missing variables or adverse selection bias. To evaluate the impact of the LTCI on family care received, this study integrates the pilot cities with the LTCI programs, involving individuals of diversities and different policy implementation years, into the treatment group, while encompassing all non-pilot cities in the control group. By comparing relevant indicators before and after policy implementation between the treatment and control groups, this paper then systematically assesses the impact of the LTCI pilot program on family care received. Based on this, the basic model of our time-varying DID is as follows:

$$Care_{ijt} = \beta_0 + \beta_1(\text{Treat}_{ij} \times Time_t) + \beta_2 \sum Z_{it} + \mu_i + \tau_t + \varepsilon_{ijt} \tag{1}$$

In Formula (1), the variables $i$, $j$, and $t$ represent the individual, city, and time, respectively. $Care_{ijt}$ refers to the outcome variable of family care. $\text{Treat}_{ij} \times Time_t$ represents the pilot variable for the LTCI, where a value of 1 indicates that the city $j$ where individual $i$ is located has implemented the LTCI pilot system in period $t$, and the individual is covered by the system; otherwise, the value is 0. $\sum Z_{it}$ represents the control variable that varies with time and individual. $\mu_i$ represents the individual fixed effect, and $\tau_t$ represents the time fixed effect. Lastly, $\varepsilon_{ijt}$ represents the standard residual terms. i = 1, 2, 3, 4, ..., N, and t = 2011, 2013, 2015, 2018.

The estimated results of model (1) are contingent upon satisfying the equilibrium trend test. This test ensures that, in the absence of policy intervention, the explanatory variables exhibit a consistent change trend in both the treatment and control groups. It is essential to verify this condition to ensure the validity of the estimated results. To achieve this objective, the present study employs the Event Study Approach (ESA) to assess the parallel trends of model (1) while analyzing the dynamic impacts of the LTCI on family care. The Eq(2) represents the model used for analyzing the dynamic effects:

$$Care_{ijt} = \beta_0 + \sum \beta_t(\text{Treat}_{ij} \times Time_t) + \beta_2 \sum Z_{ijt} + \mu_i + \tau_t + \varepsilon_{ijt} \tag{2}$$

Where, $\beta_t$ is the corresponding estimated value from 2011 to 2018. The other variables are defined as in Eq(1).

**Robustness test model.** We conducted robustness tests using two methods. Firstly, This study employed the Propensity Score Matching Difference-in-Differences (PSM-DID) model to conduct a robustness test.The Benchmark Difference-in-Differences (DID) model requires meeting the assumption of random groupings. However, when compared to an ideal

experiment, the impact of the LTCI pilots on family care received is influenced by numerous factors, making it challenging to ensure the consistency of relevant characteristics. Based on this, we employed the PSM-DID method to address the endogeneity problem arising from the potential correlation between individual characteristics and treatment/control group assignment. By controlling for covariates, we matched the treatment group and the control group to eliminate selectivity bias and more accurately evaluated the policy effect of integrating medical insurance for urban and rural residents. Before 2016, only certain regions in China independently implemented the LTCI. This study considers the national pilot cities released by the Ministry of Human Resources and Social Security in 2016 as the time of policy implementation. It constructed differential panel data using four phases of CHARLS data to empirically examine the impact of the LTCI on family care received. We used the default kernel matching for estimation in the PSM-DID model and estimated propensity scores using the Logit model. By carefully controlling for covariates, we matched individuals in the treatment group with those in the control group who have the same or similar scores. This approach helps to eliminate any selectivity bias and allows for a more accurate evaluation of the policy effect of the LTCI on family care. This paper constructed the PSM-DID regression model, as shown in Eq (3):

$$\text{Care}_{ijt}^{PSM} = \beta_0 + \beta_1(\text{Treat}_{ij} \times Time_t) + \beta_2 \sum Z_{it} + \mu_i + \tau_t + \varepsilon_{ijt} \tag{3}$$

Secondly, we adjusted the fixed effects options of the baseline model. We added city × year fixed effects, city × year effects, community fixed effects, and city × year effects with individual fixed effects to further examine the impact of the LTCI on family care received.

**Placebo test method.**   Placebo testing is a commonly used method to evaluate policy effects. It involves testing the existence of policy effects by simulating a virtual policy implementation time or processing group samples. Chinese pilot cities implementing the LTCI, like Shanghai and Qingdao, generally have severe aging populations. Moreover, these cities exhibit a higher level of economic development and a well-developed, market-oriented elderly care service system. However, it is difficult to completely rule out the impact of other unobservable factors at the individual, city, or year levels. To address this issue, we conducted placebo tests using two approaches: a virtual treatment group and the policy pilot duration.

**Heterogeneity analysis method.**   Conducting heterogeneity tests allows for a deeper examination of the relationship between the LTCI and family care received, and provides insights into the varied responses of different groups to the LTCI policies. In the present study, we conducted a group analysis focusing on older adults, considering three factors: degree of disability, marital status, and participation of retirement pension. Firstly, older adults with varying degrees of disability require different levels of daily care. When formal LTC services are insufficient, family care becomes crucial for older adults [25]. Therefore, for older adults with a high level of disability, having LTCI can help alleviate the burden of family care. It can also reduce the reliance on family care, leading to a more significant policy impact. This study, based on the research by Wang et al. focused on the need for assistance in six activities of daily living (ADL) indicators for the elderly, including eating, bathing, dressing, getting up, using the toilet, and controlling defecation and urination. The degree of disability in older adults was categorized into three levels: mild disability (difficulty in completing 1–2 ADL indicators), moderate disability (difficulty in completing 3–4 ADL indicators), and severe disability (difficulty in completing 5–6 ADL indicators). Secondly, the provision of long-term care for older adults typically follows a hierarchical compensation sequence pattern, with spouses, children, relatives, and society being the main sources of care [26]. Given that family members, particularly spouses, are the primary caregivers in long-term care, the presence of a spouse can significantly impact the level of family care received by older adults, leading to variations in the

effectiveness of policy interventions. This study aims to examine the heterogeneity of the LTCI in relation to family care by categorizing individuals based on the presence or absence of a spouse. Lastly, pension status, as a social system mandated by the state to secure the basic livelihood of retirees, represents a typical form of intergenerational public transfer. One perspective argues that pensions may reduce the level of economic support provided by children, thereby affecting their consumption and utility maximization [27].

**Mediation effect model.** This paper employed the mediating effect model to examine the mechanism by which the LTCI impacts family caregiving. To achieve this, the paper constructed mediation effect models, represented by Eqs (4)–(6):

$$C_{ijt} = \alpha_0 + \alpha_1(\text{Treat}_{ij} \times Time_t) + \alpha_2 \sum Z_{it} + \mu_i + \tau_t + \varepsilon_{ijt} \tag{4}$$

$$M_{ijt} = \gamma_0 + \gamma_1(\text{Treat}_{ij} \times Time_t) + \gamma_2 \sum Z_{it} + \mu_i + \tau_t + \varepsilon_{ijt} \tag{5}$$

$$C_{ijt} = \delta_0 + \delta_1(\text{Treat}_{ij} \times Time_t) + \delta_2 \sum Z_{it} + \delta_5 M_{ijt} + +\mu_i + \tau_t + \varepsilon_{ijt} \tag{6}$$

Let $M_{ijt}$ denote the mediating variable. Based on Eqs (4) to (6), the testing procedure is as follows: Step 1: In this step, the impact of the LTCI on family care received is examined using the basic model. Step 2: The mediating variable is added to the model as the dependent variable for testing. If the regression coefficient is significant, it indicates the presence of a mediating effect. Step 3: The mediating variables and $\text{Treat}_{ij} \times Time_t$ are separately added to the model for testing. The establishment of the mediating effect is assessed by analyzing the change in the regression coefficient.

**Ethics statement.** Ethical review and approval for this study were waived by the Institutional Review Board, because the data we used is secondary data, which is openly available to the public. All research subjects involved are anonymous.

## Results

### Descriptive statistics

Table 1 presents the descriptive statistical results for the total sample, treatment group, and control group. Regarding family care received, the mean for the entire sample was 649.873 hours/year. The treatment group had a mean of 547.107 hours/year, while the control group had a mean of 657.387 hours/year. Notably, the sample value for the treatment group was significantly lower than that of the control group. Among the processing variables, approximately 6.8% of the samples participated in the LTCI pilot with phase IV data. As for the control variables, the average age of older adults was 69.958 years. The proportion of male respondents was slightly lower than that of female, accounting for 48%. Rural household registration accounted for 60.3% of the sample. The majority of older adults had spouses, accounting for 75.1%. On average, the years of education for the sample were 3.6 years, and the proportion of healthy elderly individuals was 87.7%. A high proportion of older adults, 78%, participated in retirement pension. Upward financial support to older adults, amounting to approximately 2840.333 yuan/year, while downward financial support to their children, amounting to about 1208.358 yuan/year. In terms of age, household registration, education, pension insurance, and intergenerational economic support, the mean value of the treatment group was higher than that of the control group. However, the mean values of marriage and health status were higher in the control group compared to the treatment group.

## Main empirical results

Table 2 presents the effects of the LTCI pilot program on family care received. Model (1) displays the results without incorporating control variables, while model (2) provides the overall results with the inclusion of control variables. Models (3) and (4) present the regression findings for rural and urban samples, respectively. The overall results indicated that the LTCI decreases the amount of time older adults receiving family care by 17.2%. The regression results demonstrated statistical significance at the 5% level. The pilot policy primarily provides compensation for institutional nursing services and lowers the expenses associated with informal family care. Consequently, eligible older adults and their families tend to opt for formal care services, resulting in a reduction in the provision of informal care by family members. Hypothesis 1 is supported. This finding is also consistent with the findings of Lei et al. The results from models (3) and (4) indicated that the LTCI has a significant negative effect on family care for older adults in urban areas, but it does not have a significant impact on rural family care. Firstly, the pilot cities for the LTCI primarily consist of employed individuals, workers, and urban residents. These specific target groups predominantly reside in urban areas, making them more susceptible to the influence of LTCI policies compared to rural areas. Additionally, rural residents generally have lower income levels and have a stronger adherence to the traditional family nursing concept, resulting in family care being a crucial method of long-term care for older adults in these areas. Despite the implementation of the LTCI providing coverage for disability risk and lessening the financial burden of caring for rural families, it is unable to alter the fundamental principle of intergenerational support in

**Table 2. Main regression results.**

| Variables | (1) | (2) | (3) | (4) |
|---|---|---|---|---|
| $\text{Treat}_{ij} \times \text{Time}_t$ | -0.135** | -0.172** | -0.128 | -0.197** |
| | (0.081) | (0.099) | (0.101) | (0.197) |
| Age | | -0.092*** | -0.088*** | -0.096*** |
| | | (0.004) | (0.005) | (0.006) |
| Gender | | 0.408*** | 0.260*** | 0.627*** |
| | | (0.051) | (0.066) | (0.082) |
| Household registration | | -0.083* | —— | —— |
| | | (0.050) | | |
| Marriage status | | -0.773*** | -0.728*** | -0.846*** |
| | | (0.057) | (0.072) | (0.095) |
| Education level | | 0.031*** | 0.038*** | 0.025** |
| | | (0.008) | (0.011) | (0.012) |
| Health status | | 0.338*** | 0.346*** | 0.315*** |
| | | (0.073) | (0.092) | (0.120) |
| Endowment insurance | | 0.134*** | 0.133* | 0.136 |
| | | (0.058) | (0.075) | (0.092) |
| _Cons | 1.363*** | 7.136*** | 6.978*** | 6.848*** |
| | (0.022) | (0.322) | (0.384) | (0.571) |
| Time fixed effects | YES | YES | YES | YES |
| Individual fixed effects | YES | YES | YES | YES |
| N | 18948 | 18948 | 11426 | 7522 |

Note

* $p<0.1$

** $p<0.05$

*** $p<0.01$.

## Parallel trend and dynamic effects

An essential requirement for evaluating the policy effect using the time-varying DID model is that, prior to the implementation of the LTCI, the development trend of family care time should be consistent between the treatment group and the control group, with no systematic differences between them. The divergence between the two groups should manifest itself following the implementation of the policy. In light of this, the present study employed the Event Study Approach to test the dynamic impact and validate the parallel trend hypothesis. After accounting for individual and time fixed effects, the findings of the parallel trend analysis for family care before and after the introduction of LTCI are presented in Fig 2. In the two pre-policy implementation periods, all regression results were found to be statistically insignificant, with similar values. However, following the implementation of the policy, a significant decrease in the trend of family care was observed. This change in trend, both before and after the policy, aligned with the parallel trend analysis and provided further support for the robustness of the baseline regression results.

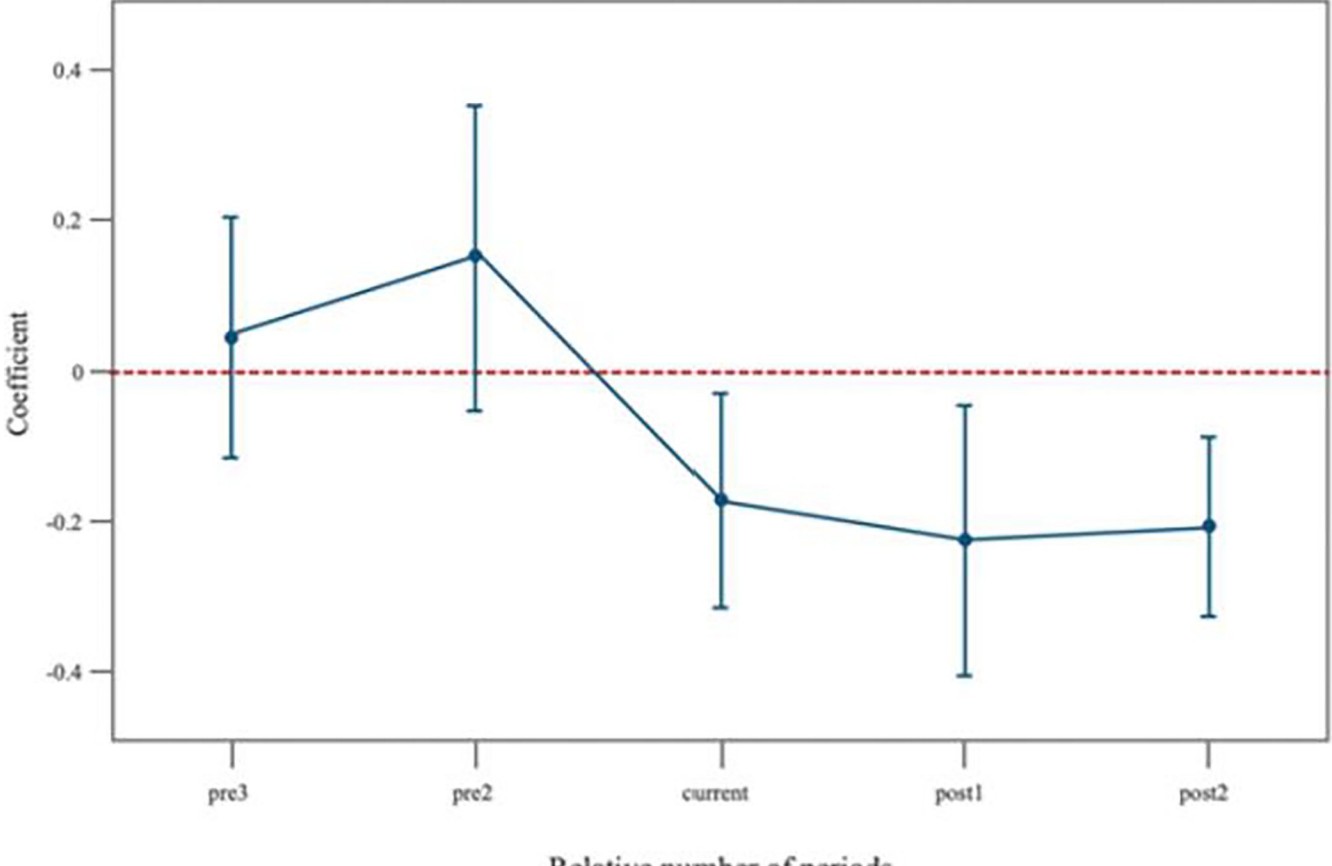

**Fig 2. Dynamic effects of the LTCI on family care received (95%CI).**

**Table 3. Balance test results for the PSM-DID model.**

| Variable | Unmatched Matched | Mean | | %reduct | | t-test | |
|---|---|---|---|---|---|---|---|
| | | Treated | Control | %bias | \|bias\| | t | p>\|t\| |
| Age | Unmatched | 72.744 | 70.368 | 35.0 | 91.3 | 10.37 | 0.000 |
| | Matched | 72.744 | 72.537 | 3.1 | | 0.69 | 0.490 |
| Gender | Unmatched | 0.419 | 0.432 | -2.8 | 17.1 | -0.83 | 0.408 |
| | Matched | 0.419 | 0.408 | 2.3 | | 0.51 | 0.613 |
| Education level | Unmatched | 3.689 | 3.387 | 8.8 | 46.9 | 2.88 | 0.004 |
| | Matched | 3.689 | 3.529 | 4.7 | | 0.98 | 0.326 |
| Household registration | Unmatched | 0.753 | 0.599 | 33.2 | 85.3 | 9.50 | 0.000 |
| | Matched | 0.753 | 0.775 | -4.9 | | -1.17 | 0.241 |
| Health status | Unmatched | 0.851 | 0.878 | -7.9 | 13.0 | -2.48 | 0.013 |
| | Matched | 0.851 | 0.875 | -6.9 | | -1.52 | 0.130 |
| Marriage status | Unmatched | 0.715 | 0.734 | -4.3 | 8.5 | -1.30 | 0.195 |
| | Matched | 0.715 | 0.732 | -3.9 | | -0.86 | 0.390 |
| Endowment insurance | Unmatched | 0.799 | 0.779 | 4.9 | 100.0 | 1.46 | 0.143 |
| | Matched | 0.799 | 0.799 | 0.0 | | 0.00 | 1.000 |

## Robustness tests

The balance test of the PSM-DID model is presented in Table 3. Before the matching, there were significant imbalances between the treatment and control groups in some variables (such as age and household registration). After matching, the standardized biases of all covariates were below 10%, and the t-test results did not reject the null hypothesis of no systematic bias between the treatment and control groups. In addition, after performing propensity score matching, there were 495 samples in the control group that were not within the common range of values. However, all samples in the treatment group were within the common range. As a result, only a small number of samples (approximately 2.6%) were lost during the propensity score matching process. This indicates that all variables have passed the balance test. Table 4 reports the results of two robustness tests. Whether it is the PSM-DID test or the adjusted fixed effects model, the LTCI significantly reduces family care received, further supporting the robustness of the baseline regression results.

**Table 4. Results of the robustness tests.**

| Test method/Variable | Coeff | Std. Err. | N |
|---|---|---|---|
| (1) PSM-DID test | | | |
| Family care | -0.510** | 0.214 | 13652 |
| (2) Adjusting fixed effects | | | |
| Added city × year fixed effect | -0.163** | 0.229 | 10495 |
| Increased city × year, community fixed effect | -0.139** | 0.195 | 8937 |
| Increase city × year, individual fixed effect | -0.103** | 0.199 | 12203 |

*Notes*. The PSM-DID model results were estimated using a Logit model to calculate propensity scores and employing default kernel matching for estimation.

* p<0.1

** p<0.05

*** p<0.01.

## Placebo tests

This study utilized data from the China Health and Retirement Longitudinal Study (CHARLS) for the period of 2011 to 2018 to examine the impact of the LTCI pilot program on family care received. The analysis does not include data beyond 2018, as it is not influenced by the LTCI policy pilot after this period. In 2020, the National Medical Insurance Bureau and the Ministry of Finance designated 13 additional regions, including Beijing, Tianjin, Fuzhou, Kaifeng, and Kunming, as the second group of pilot cities for the program. Based on this, the present study established a virtual treatment group (denoted as "$Treat_i$") consisting of the second batch of pilot cities, while the remaining cities constitute the control group. The variable $Treat_{ij} \times Time_t$ is assumed to have no substantial influence on family care received. The test results are presented in Table 6, which shows that the regression coefficient of $Treat_{ij} \times Time_t$ is statistically insignificant at the 10% level. This suggests that the estimation results remain unaffected by individual and time variations, thereby confirming the robustness of the baseline regression findings. The second method involved using a virtual implementation time for the policy. In the present study, a time-varying difference-in-differences (DID) model was employed to examine the impact of the LTCI on family care received. As it is not feasible to establish a uniform treatment time, we excluded pilot cities before 2016, considering that most LTCI pilots were conducted after this year. The treatment group consisted of pilot cities after 2016, while the control group comprised other cities. We assumed that the policy implementation time for the pilot cities was either 2012 or 2014. For the analysis, we utilized pre-policy implementation data from 2011 and post-policy performance data from 2013, 2015, and 2018 when assuming 2012 as the policy time. In the case of assuming 2014 as the policy implementation time, the pre-implementation period included data from 2011 and 2013, while the post-implementation period involved data from 2015 and 2018. The regression findings presented in Table 5 indicate that the estimated effects of the virtual policy pilot in 2012 and 2014, and were not statistically significant.

## Heterogeneity analysis

Table 6 presents the results of the heterogeneity tests for the three groups. Firstly, it is found that the LTCI has a significant negative impact on family care received for the moderately and severely disabled elderly, but it does not have a considerable effect on family care received for

**Table 5. Placebo tests.**

| Variables | (1) | (2) | (3) |
|---|---|---|---|
| | **Virtual treatment group** | **Virtual 2012 as policy time** | **Virtual 2014 as policy time** |
| $Treat_{ij} \times Time_t$ | 1.073 | -0.218 | -1.139 |
| | (0.707) | (0.247) | (0.718) |
| _Cons | -7.101*** | 6.534*** | -11.659 |
| | (0.412) | (1.767) | (1.869) |
| Control variables | YES | YES | YES |
| Time fixed effects | YES | YES | YES |
| Individual fixed effects | YES | YES | YES |
| N | 18948 | 14051 | 14051 |

Note

* p<0.1

** p<0.05

*** p<0.01.

**Table 6. Heterogeneity analysis.**

| Variables | Degree of disability | | | | Health status | | Endowment insurance | |
|---|---|---|---|---|---|---|---|---|
| | (1) Health | (2) Mild disability | (3) Moderate disability | (4) Severe disability | (5) With spouse | (6) Without spouse | (7) With | (8) None |
| $Treat_{ij} \times Time_t$ | -1.154 | -0.152** | -0.294** | -0.242** | -0.227** | -0.043 | -0.062** | -0.118** |
| | (0.110) | (0.368) | (0.257) | (0.351) | (0.116) | (0.184) | (0.100) | (0.202) |
| _Cons | 7.640*** | 6.416*** | 3.193*** | 3.379*** | 5.917*** | 9.462*** | 7.348*** | 6.324*** |
| | (0.338) | (0.858) | (1.242) | (1.309) | (0.353) | (0.541) | (0.313) | (0.588) |
| Control variables | YES | YES | YES | YES | YES | YES | YES | YES |
| Time fixed effects | YES | YES | YES | YES | YES | YES | YES | YES |
| Individual fixed effects | YES | YES | YES | YES | YES | YES | YES | YES |
| N | 13078 | 1402 | 246 | 183 | 14226 | 4722 | 14766 | 4160 |

Note

* p<0.1

** p<0.05

*** p<0.01.

non-disabled older adults. This indicates that the LTCI enhances the availability of socialized care services, and the severely disabled elderly are more inclined to opt for specialized care services as a viable substitute for family care. Secondly, the LTCI has no significant impact on the level of family care received for older adults with spouses but does not significantly affect the group without spouses. Spouses represent the first choice for informal family care. In comparison to older adults without spouses, when older adults with spouses receive formal LTC services, they tend to decrease family care services for their other family members. Older adults without spouses, irrespective of their access to LTC services, need to provide some family care, which is less influenced by the LTCI. Thirdly, the LTCI has a greater crowding-out effect on family care for older adults who do not have spouses. As a formal social security system, the LTCI has a more pronounced impact on older adults who lack formal social support. According to Fan, the LTCI services can provide temporary relief from the burden of family care for individuals who lack formal social support [28]. Therefore, Hypothesis 2 is supported.

## Mechanism analysis

Traditional family responsibility theory indicates a strong and unique bond between blood relatives. As individuals age, it is expected that adult children will assume the responsibility of providing both upward financial support and daily care. Altruism theory suggests that within a family, the altruistic motive is the key to achieving optimal resource allocation, thereby maximizing the collective interests of the family as a whole, rather than individual interests [29]. In the context of the massive rural-urban migration, escalating living expenses, and a growing employment prospects for women, there has been a gradual increase in the number of rural female laborers seeking work opportunities in urban areas. This trend has also led to the emergence of dual-income families in urban areas. Consequently, adult children in these families often find themselves caught between the demands for upward economic support and the need for physical family care. On one hand, some adult children enter labor force to secure an income and subsequently offer financial assistance to older adults to partially substitute for physical familial caregiving. On the other hand, adult children may opt to decrease their working hours and corresponding income while increasing the time allocated to family care. Currently, within the context of shrinking family size and dual-income families, substituting family care with economic support is becoming more evident. LTCI, under such context,

gurantees the provision of formal caregiving to older adults while diminishing their financial reliance on adult children, therefore alleviating adult children's care burden. In other words, LTCI provides an approach to decrease the extent of informal caregiving, and substitutes it with a portion of the financial support previously provided by adult children.

The traditional Chinese approach of providing elderly care centers around intergenerational support within families, which encompasses three main aspects: economic support, physical caregiving support, and emotional support. Within the family unit, intergenerational relationships serve as a model of reciprocal exchange. For instance, parents raise and financially support their children along the life course with an expectation of receiving informal care in return for old age. However, LTCI, as a formal social security system, offers institutionalized protection for disabled older adults and reduces their reliance on informal family care, which may diminish their motivations to provide downward support to adult children at the first place for the traditional intergenerational exchanges that previously existed.

Table 2 presents the findings that demonstrate a significant reduction in informal family care as a result of the LTCI, indicating the validity of the first step test. Building upon these results, this study proceeded to conduct the second and third step tests. The outcomes of the second step test, as shown in Table 7, indicated a significant decrease in both "upward" and "downward" financial support due to the LTCI. This suggests that the LTCI weakens the motivation for intergenerational exchange among older adults and alleviates the economic burden of family care for adult children. Furthermore, the results of the third step test revealed a significant improvement in family care as a result of the significant reduction in "upward" and "downward" financial support. In summary, the LTCI diminishes the intergenerational financial support between the elderly parents and their adult children, consequently reducing the motivation for family care. These findings confirm the validity of hypotheses 3 and 4.

## Further analysis

This study discovered that the LTCI has a substantial impact on the reduction of family care. While informal family care in Chinese families is struggling to meet the LTC needs of the

**Table 7. Analysis of influence mechanism.**

| Variables | The second step test | | The third step test | |
|---|---|---|---|---|
| | (1)<br>"Upward" financial support | (2)<br>"Downward" financial support | (3)<br>Family care | (4)<br>Family care |
| $Treat_{ij} \times Time_t$ | -0.098***<br>(0.035) | -0.003***<br>(0.028) | -0.086<br>(0.094) | -0.171*<br>(0.098) |
| "Upward" financial support | | | 0.882***<br>(0.022) | |
| "Downward" financial support. | | | | 0.529***<br>(0.029) |
| _Cons | 0.656***<br>(0.115) | 0.038***<br>(0.092) | 6.558***<br>(0.305) | 7.116***<br>(0.318) |
| Control variables | YES | YES | YES | YES |
| Time fixed effects | YES | YES | YES | YES |
| Individual fixed effects | YES | YES | YES | YES |
| N | 18948 | 18948 | 18948 | 18948 |

Not
.* $p<0.1$
** $p<0.05$
*** $p<0.01$.

**Table 8. Impact of LTCI on the health of older adults.**

| Variables | (1) | (2) | (3) | (4) |
|---|---|---|---|---|
| | **Chronic diseases** | **ADLs** | **CES-D** | **Self-rated health** |
| $\text{Treat}_{ij} \times \text{Time}_t$ | -0.029 | 0.006 | -1.063*** | 0.139 |
| | (0.072) | (0.011) | (0.355) | (0.043) |
| _Cons | 5.026*** | 1.416*** | 28.647*** | 3.008*** |
| | (0.529) | (0.035) | (3.298) | (0.129) |
| Control variables | YES | YES | YES | YES |
| Time fixed effects | YES | YES | YES | YES |
| Individual fixed effects | YES | YES | YES | YES |
| N | 18948 | 18948 | 11277 | 18948 |

*Notes.*The analysis included a range of chronic diseases such as hypertension, diabetes, heart disease, stroke, cancer, arthritis, Parkinson's disease, and other non-communicable chronic diseases. The ADLs index encompassed six essential activities of daily living: eating, bathing, dressing, getting up, using the toilet, and controlling defecation and urination. Mental health was assessed using the International Center for Epidemiological Research Self-Rating Scale for Depression (CES-D). Self-rated health was evaluated through the question "How do you perceive your overall health?".

* $p < 0.1$

** $p < 0.05$

*** $p < 0.01$.

growing number of disabled older adults. LTCI helps alleviate the care burdens of adult children and gurantees the provision of specialized care services for older adults. However, does this crowding-out effect actually result in improved care for older adults? In essence, the question arises as to whether the crowding-out effect of the LTCI on family care compromises the health of older adults. This paper further examined the influence of the LTCI on the health of older adults across four key health measures: the prevalence of chronic diseases, mental health, ADLs, and self-rated health. Findings presented in Table 8 revealed that the LTCI has a significant positive effect on improving older adult's mental health and reducing their depression. However, no significant impact was observed between the LTCI and the number of chronic diseases, ADL indicators, and self-rated health. Consequently, it is inferred that the crowding-out effect of the LTCI on family care does not compromise the physical health of older adults. Moreover, the LTCI demonstrates a beneficial impact on older adult's mental health, which aligns with the policy objectives.

## Discussion

Based on the pilot program of the LTCI in China, this study assessed the impact of the LTCI on family care received by older adults, with the help of employing various robustness tests, including PSM-DID, Event Study Approach, and the placebo tests. This study also explored the heterogeneity effects and the underlying mechanisms between the relationship to offer insights for optimizing the LTCI pilot program. In addition, this study further examined whether the crowding-out effect between the LTCI and informal family care jeopardizes older adult's health. The research findings of this study are summarized as follows.

Firstly, the LTCI significantly reduced family care for older adults, and its impact on older adults in urban areas was higher than that in rural areas, which is consistent with the previous research [14,16]. At present, family care remains the primary source of old-age care and the primary means of long-term care for disabled older adults. However, rapid population aging and the increasing number of disabled older adults have posited challenges for both Chinese informal caregivers and China's economic and social development. The LTCI system has emerged as a crucial measure for alleviating the pressure on family care and meeting older

adult's LTC needs [30]. However, the coverage of the LTCI in pilot cities mainly targets urban workers because urban areas share the convenience of both the institutional and family care resources. This makes urban residents more susceptible to the policy effects as compared to rural residents. However, this heterosity may disappear as it gradually includes both urban and rural residents when the pilot scope expands.

Secondly, the LTCI has stronger effect on family care services for older adults with moderate to severe disabilities, which is also consistent with the prior research [19,31]. According to Zhu & He, the LTCI gives priority to older adults with severe disabilities. These individuals are more likely to opt for formal nursing services, leading to a more noticeable replacement of family care received. Moreover, the LTCI has stronger replacement effect on family care received among older adults with spouses. When older adults with spouses receive formal LTC services, it tends to replace family care provided by other family members, amplifying the impact of the policy. The LTCI also has observed replacement effect on family care received among older adults without spouses, which is in line with the findings of Kim & Lim [32]. As a formal social security program, pensions diminish the economic support provided by adult children, aiming to maintain consumption and maximize utility.

Thirdly, the LTCI diminishes the provision of family care by decreasing both "upward" and "downward" financial assistance, aligning with the research findings of Zhu [17]. This suggests that the LTCI plays a beneficial role in alleviating the intergenerational support burden on families, offering institutionalized societal support for family-based caregiving, and enhancing overall social welfare. The relationship between formal and informal care has received widespread attention. As a formal care system, the LTCI helps alleviate the burden on families and promotes the development of formal care. This fully demonstrates the complementary relationship between the two. Building formal care system supporting older adults on the foundation of informal care within families is beneficial for ensuring the older adults' LTC needs.

The study has several limitations. Firstly, the CHARLS data used in this paper does not include some pilot cities due to data inavailability. However, the sample size of the available data is beyond sufficient for our analytical purposes. Secondly, regional differences were not sufficiently addressed in this study (East, Middle, West, and Northeast China). We thus call for future studies that address regional differences to add to the findings of this study. Thirdly, it should be noted that the CHARLS dataset does not provide information on the overall response rate, which could potentially affect the findings of this study. However, it is fortunate that the data processing results indicate that the response rates for various questions relevant to this study in the CHARLS dataset exceed 90%, thus ensuring the reliability of the research findings.

## Conclusions and policy implications

Using panel data of CHARLS from 2011 to 2018, this study assessed the impact of the LTCI pilot program on family care received by older adults. To analyze this effect, time-varying DID models were estimated. The results revealed several important findings. Firstly, the LTCI significantly diminished the amount of care provided by family members, leading to a 17.2% reduction in family care received by the Chinese older adults in the pilot areas. This reduction effectively alleviates the burden of family care. Secondly, the impact of the LTCI on family care varies among different groups, with a more pronounced effect observed in urban areas, among those with severe disabilities, married individuals, and elderly individuals without retirement pension. Thirdly, the LTCI achieved its reduction in family care by decreasing intergenerational financial support. Most importantly, the reduction of family care caused by the LTCI does not have a detrimental impact on older adult's physical health. Meanshile, it was found to

be beneficial in terms of reducing depression levels and improving overall mental health for older adults.

Based on its social advantage of alleviating family care burdens and health advantages of promoting older adult's physical and mental well-being, we propose the following policy implications for establishing a national LTCI system in China: Firstly, this study suggests that the LTCI significantly reduces the burden of family care and provides specialized LTC services for older adults. As the role of family care is gradually weakening, the government should continue to expand the scope of the LTCI pilot programs to provide institutionalized support for older adults. Secondly, this study demonstrates that the impact of the LTCI is pronounced in urban regions as compared to rural areas. This objectively reflects the urgent need for long-term care services to be established and delivered in these areas. Thirdly, the LTCI falls under formal care, while family care is considered informal caregiving. Therefore, it is important to integrate the two appropriately. The implementation of the LTCI should not completely replace family care, as each caregiving approach has its own advantages and disadvantages. Promoting a balanced development between the two and providing more choices for older adults and their families is an inevitable trend for the LTCI's future development.

## Acknowledgments

We express our gratitude for the data support provided by the publicly available database, China Health and Retirement Longitudinal Study (CHARLS).

## Author Contributions

**Methodology:** Jing Liu.

**Visualization:** Jing Liu.

**Writing – original draft:** Lianjie Wang.

**Writing – review & editing:** Jing Liu.

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
