## [Decision Letter · Decision Letter 0]

8 Nov 2023

PONE-D-23-25847The Impact of Long-term Care Insurance on Family Care for Older Adults: Evidence from a Quasi-Natural Experiment in ChinaPLOS ONE

Dear Dr. wang,

Thank you for submitting your manuscript to PLOS ONE. After careful consideration, we feel that it has merit but does not fully meet PLOS ONE’s publication criteria as it currently stands. Therefore, we invite you to submit a revised version of the manuscript that addresses the points raised during the review process.

We look forward to receiving your revised manuscript.

Kind regards,

Yuxia Wang

Academic Editor

PLOS ONE

Additional Editor Comments:

Dear Authors,

We received two reviews of your manuscript. While Reviewer #2 recommended rejection of this work, I would like to reconsider this manuscript after a major revision hoping to see substantial improvement. During revision, please pay attention to the suggestion and comments of the Reviewer #2. Please note that your revised version will be further assessed by external reviewers.

Reviewers' comments:

Reviewer's Responses to Questions

**Comments to the Author**

1. Is the manuscript technically sound, and do the data support the conclusions?

Reviewer #1: No

Reviewer #2: Yes

2. Has the statistical analysis been performed appropriately and rigorously? 

Reviewer #1: No

Reviewer #2: Yes

3. Have the authors made all data underlying the findings in their manuscript fully available?

Reviewer #1: Yes

Reviewer #2: Yes

4. Is the manuscript presented in an intelligible fashion and written in standard English?

Reviewer #1: Yes

Reviewer #2: No

5. Review Comments to the Author

Reviewer #1: This study examines how the introduction of long-term care insurance in China has affected the amount of family care provided to older adults using a difference-in-differences analysis approach, taking into account the heterogeneity of the subjects. The results show that the introduction of long-term care insurance leads to a decrease in family care. They also show that the impact of insurance varies depending on the attributes of the target population. The authors demonstrate the effectiveness of long-term care insurance.

This paper provides important insights into how to support family caregiving for older adults in an aging society. The hypothesis is clearly presented, the analysis is well designed, and the results are interesting. However, as interesting as the study is, it is very disappointing that the authors disregard the journal's author guidelines. In addition, the validity of the data used is unclear and there seems to be insufficient consideration of limitations. I hope you will take the following points into consideration.

Major comments

1. The author guidelines contribute to readability, so the authors should be divided accordingly into Introduction, Materials and Methods, Results, Discussion, and Conclusions. I understand that Results and Discussion are sometimes written together in economics and policy studies, but at least in 4.4.2, 4.5, and Table 8, analyses not described in Methods should not be started in the Results and Discussion sections. The Methods section should include a description of the analyses for all robustness checks and heterogeneity analyses, as well as the analyses that accompany them (e.g., balance test). The Vancouver method should also be used for references in accordance with the guidelines.

2. 2.1 Policy background: Please consider adding a description of the system (e.g., delivery system, support system, etc.) for medical care and long-term care for older people in China before the intervention begins. My concern is that when financial support is provided through long-term care insurance, residents will not be able to use them if the local service delivery system is not adequate. Also, the medical care delivery system could complement long-term care services.

3. 2.1 Policy background: Please consider specifying the differences between medical care and long-term care in the Chinese system. In particular, it would be helpful to explain the differences in financing, payment, service delivery systems, and service content.

4. Figure 1: It would be easier to understand if you could visualize which regions are starting to introduce the system and from which survey year. I am not familiar with Chinese place names, so I have difficulty understanding the text and figures; I suggest numbering the pilot areas and indicating the place names with notes. Also, please let me confirm what is the difference between the pilots by the local government and the officially announced 15 pilots.

5. 2.2 Literature Review: Prior studies have been properly reviewed, but it would help the reader's understanding by specifying which country the study is from.

6. 3.1 Data: CHARLS data is insufficiently detailed. At the least, the survey's subject and target sampling methods, survey methodology, geographic areas included, response rates, and dropout rates are important for bias and interpretation of the results. Please consider providing explanations in the text and supporting materials or citing articles that describe them.

7. please specify how many subjects were excluded in each of the two stages of the selection process.

8. Were there any missing data? If so, how were they addressed?

9. 3.3.1: Individuals are nested in cities. Therefore, I am concerned that each observation is not independent over time and that there is a city-level autocorrelation. For example, have you considered using city cluster standard errors or using city random effects?

10. 3.3.2: Did the authors use "nearest neighbor matching within caliper"? Please provide more clarification on the matching method.

11. 3.3: Did they use linear regression for all analyses?

12. Table 1: Considering the DID, it would be easier to understand the changes if you show the mean values before and after the intervention for each group, respectively.

13. Figure 2: Does "current" on the horizontal axis indicate the time of intervention? How is it taken into account in this analysis if there is only one time point before the intervention or one time point after the intervention?

14. 4.4.1: Was the overlap of the propensity score values sufficient?

15.Line 608: What do the authors consider the validity of the common shocks assumption for DID in this study?

16. Policy Implications: We should limit ourselves to suggestions that can be made based on the results analyzed in this study. Some of the suggestions seem to be overstated.

Minor comments

1. 3.3.3: Equations 7 and 8 are missing.

2. Table 4: seems to contain Chinese characters.

3. Line 594: Table 9 is missing.

Reviewer #2: 1.There is much room for modification in the language, typesetting and grammatical methods of the article.

2.Please strengthen introduction section to highlight the value of this study. Besides, Each paragraph in the introduction seems to be unrelated to the previous paragraph.

3.There is an expression mistake, not '家庭照料' but 'family care'.

4.Regarding sample selection, to enhance the reliability of research conclusions, the author needs to accurately match pilot samples.

6. PLOS authors have the option to publish the peer review history of their article (what does this mean?). If published, this will include your full peer review and any attached files.

Reviewer #1: No

Reviewer #2: No

---

## [Author Response · Author response to Decision Letter 0]

12 Dec 2023

Response to Reviewers

Dear Editor and Reviewers,

We would like to express our gratitude for your professional suggestions and constructive comments, which are essential for improving the quality of our research paper. We have carefully considered and addressed each point raised by the reviewers, and we believe that our revised manuscript is now significantly improved. Please find below a detailed response to each of the reviewers' comments.

1. Please ensure that your manuscript meets PLOS ONE's style requirements, including those for file naming. The PLOS ONE style templates can be found at https://journals.plos.org/plosone/s/file?id=wjVg/PLOSOne_formatting_sample_main_body.pdf
https://journals.plos.org/plosone/s/file?id=ba62/PLOSOne_formatting

_sample_title_authors_affiliations.pdf.

Response:

Thank you for the suggestions provided by the journal. We have made modifications to the format of this article according to the requirements of the two documents, making the manuscript more in line with the journal’s requirements.

Response:

Thank you for your reminder. We have carefully checked and corrected the funding detail.

Response:

Thank you so much for this kind reminder. The data we used in this article is secondary data, which is openly available to the public. If you need anything else from us, we will be more than happy to provide

Response:

Thank you for your comment. We have added the "Ethics statement" in the "Methods" section of this paper on page 11.

Response:

Thank you so much. Figure 1 was drawn by the authors, therefore, the authors own the copyright of Figure 1. The authors grant permission for the open-access journal PLOS ONE to publish Figure 1 under the Creative Commons Attribution License (CCAL) CC BY 4.0.

Reviewer #1:

This study examines how the introduction of long-term care insurance in China has affected the amount of family care provided to older adults using a difference-in-differences analysis approach, taking into account the heterogeneity of the subjects. The results show that the introduction of long-term care insurance leads to a decrease in family care. They also show that the impact of insurance varies depending on the attributes of the target population. The authors demonstrate the effectiveness of long-term care insurance.

This paper provides important insights into how to support family caregiving for older adults in an aging society. The hypothesis is clearly presented, the analysis is well designed, and the results are interesting. However, as interesting as the study is, it is very disappointing that the authors disregard the journal's author guidelines. In addition, the validity of the data used is unclear and there seems to be insufficient consideration of limitations. I hope you will take the following points into consideration.

Major comments

1. The author guidelines contribute to readability, so the authors should be divided accordingly into Introduction, Materials and Methods, Results, Discussion, and Conclusions. I understand that Results and Discussion are sometimes written together in economics and policy studies, but at least in 4.4.2, 4.5, and Table 8, analyses not described in Methods should not be started in the Results and Discussion sections. The Methods section should include a description of the analyses for all robustness checks and heterogeneity analyses, as well as the analyses that accompany them (e.g., balance test). The Vancouver method should also be used for references in accordance with the guidelines.

Response: 

Thank you for your suggestions. We have revised the manuscript as follows:

Firstly, according to the author's guidelines, we have revised the article structure into following 6 sections: Introduction, Policy background and research hypothesis, Materials and methods, Results, Discussion, Conclusions and policy implications. Based on the original manuscript, we also added a section named "Policy background and research hypotheses" to introduce more details of the pilot long-term care insurance policies in China. We hope that’ll help provide more background information for readers.

Secondly, we have revised the structure of the methods, results and discussion sections. In the "methods" section, we have presented an overview of all the primary research methods conducted in this study. We have consolidated all the empirical analysis in the "results" section. Through these modifications, we have ensured that the "methods," "discussion," and "results" sections are more in line with the basic requirements of the journal.

2. 2.1 Policy background: Please consider adding a description of the system (e.g., delivery system, support system, etc.) for medical care and long-term care for older people in China before the intervention begins. My concern is that when financial support is provided through long-term care insurance, residents will not be able to use them if the local service delivery system is not adequate. Also, the medical care delivery system could complement long-term care services.

Response:

Thank you so much for this valuable comments. We have rewritten the entire "policy background" section. Please see from page 3 to page 5. In the revised version, we introduced the system in two sub-sections: the first primarily analyzes the differences between China's medical care and long-term care insurance, while the second provides a brief overview of the development and basic situation of long-term care insurance. In China, Long-term care insurance is composed of separately established healthcare institutions that introduce market entities such as nursing homes and medical institutions to provide services. Based on our research in various regions, it is observed that older adults who qualify for long-term care insurance can generally receive comprehensive and qualified nursing services. In addition, medical care covers expenses related to the prevention and treatment of chronic diseases and illnesses in older adults, while long-term care insurance primarily provides service support for disabled older adults. 

3. 2.1 Policy background: Please consider specifying the differences between medical care and long-term care in the Chinese system. In particular, it would be helpful to explain the differences in financing, payment, service delivery systems, and service content.

Response: 

Thank you for your suggestion. We have discussed the differences between China’s medical insurance and long-term care insurance in the policy background section on page 3 and 4.

4. Figure 1: It would be easier to understand if you could visualize which regions are starting to introduce the system and from which survey year. I am not familiar with Chinese place names, so I have difficulty understanding the text and figures; I suggest numbering the pilot areas and indicating the place names with notes. Also, please let me confirm what is the difference between the pilots by the local government and the officially announced 15 pilots.

Response:

Thank you for your suggestions. I have made the following revisions to the manuscript:

Firstly, Figure 1 presents the basic situation of the pilot program for the long-term care insurance in China. The subsequent research in the manuscript is based on the treatment groups annotated in the CHARLS data from the regions depicted in Figure 1. We totally agree with you that adding the specific policy implementation year for each pilot city will make more sense for the readers while reading this paper. We have added detailed policy implementation years for all the pilot cities in the "Note" section of Figure 1. 

Secondly, the officially announced 15 pilot cities are national-level pilot cities that have been determined by the central government through the issued documents. They are part of a formal national policy promoted by the central government. On the other hand, local government pilots are initiated by the local governments themselves and are local policies undertaken to address the long-term care needs of older adults. The general pattern of institutional development in China is to first select representative regions for regional pilots, gradually expand the scope of the pilot program, and eventually implement it nationwide.

We really appreciated the suggestion of numbering the pilot cities to help readers distinguish different cities. But after careful consideration, we did not number the pilot cities in this revised version, instead, we added the specific policy implementation year for each city to help readers distinguish the pilot cities.

5. 2.2 Literature Review: Prior studies have been properly reviewed, but it would help the reader's understanding by specifying which country the study is from.

Response:

Thank you for your suggestion. It is indeed important to acknowledge that different scholars’ perspectives are based on different countries and regions. We have made efforts to indicate the countries associated with different viewpoints as much as possible, so that readers can have a clearer understanding of this.

6. 3.1 Data: CHARLS data is insufficiently detailed. At the least, the survey's subject and target sampling methods, survey methodology, geographic areas included, response rates, and dropout rates are important for bias and interpretation of the results. Please consider providing explanations in the text and supporting materials or citing articles that describe them.

Response:

Thank you for your comments. Our description of the data source is primarily based on the official website of CHARLS, which can be found at http://charls.pku.edu.cn/gy/gyxm.htm. In response to your suggestion, we have added additional detailed information in the data section (page 8) to provide more detailed information about the data. This ensures that our data contains sufficient details.

The followings are the data details we added in this revision:

(1) Sponsor of the data Project: Peking University National Development Research Institute, Peking University China Social Science Survey Center, and Peking University Youth League Committee.

(2) Start time and frequency of follow-up surveys: The survey started in 2011, and three follow-up surveys were conducted in 2013, 2015, and 2018.

(3) Coverage: The survey covers 28 provinces in China and includes nearly 20,000 respondents from 450 communities nationwide (Excluding Hong Kong, Macau and Taiwan Province).

(4) Main methodology: The questionnaire design drew on international experiences, including the Health and Retirement Study (HRS) in the United States, the English Longitudinal Study of Aging (ELSA), and the Survey of Health, Aging, and Retirement in Europe (SHARE), among others. The project employed a multi-stage sampling approach, with the use of probability proportional to size (PPS) sampling at both the county/district and village levels. CHARLS pioneered the use of electronic mapping software (CHARLS-GIS) technology to create village-level sampling frames using a mapping method.

(5) We also added the detailed description of the two reasons, process, and results of using CHARLS in this study.

Unfortunately, CHARLS did not disclose response rates, therefore we were not able to talk about it in the article. But we made sure that we include detailed sample and variable information in the descriptive statistics section. Thank you for your understanding and support.

7. please specify how many subjects were excluded in each of the two stages of the selection process.

Response:

Thank you for your suggestion. I have added the number of lost samples during the two stages of processing on page 8.

8. Were there any missing data? If so, how were they addressed?

Response:

Thank you for your comments. Yes, there were missing data in the variables "Upward" economic support and "Downward" economic support. Since these two variables are continuous, we used the interpolation method to handle missing values by predicting them based on neighboring observations. Taking your advice into consideration, we have added supplementary explanations in the note section of Table (page 13).

9. 3.3.1: Individuals are nested in cities. Therefore, I am concerned that each observation is not independent over time and that there is a city-level autocorrelation. For example, have you considered using city cluster standard errors or using city random effects?

Response:

Thank you for your comment. We greatly agree with your point of view. In the basic regression, we only controlled for time and individual fixed effects, without considering city random effects. Therefore, we have made modifications in the "robustness test" section. In this section, we have added city and province fixed effects, as well as some interaction terms. We hope that this approach will help address the issue.

10. 3.3.2: Did the authors use "nearest neighbor matching within caliper"? Please provide more clarification on the matching method.

Response:

Thank you for your suggestion. The standalone PSM method provides multiple matching methods, and the PSM-DID model typically uses default kernel matching for estimation. In accordance with your suggestion, we have made modifications to the corresponding section and explained the method we used.

11. 3.3: Did they use linear regression for all analyses?

Response:

Thank you for your comment. The dependent variable in this study is a continuous variable, and the data used is panel data. Therefore, the main research method employed is the linear regression method (“xtreg”). In the mechanism analysis, intergenerational economic support is also a continuous variable. Therefore, it is mostly analyzed using the linear regression method, unless otherwise specified.

12. Table 1: Considering the DID, it would be easier to understand the changes if you show the mean values before and after the intervention for each group, respectively.

Response:

Thank you for your comment. We have made supplementary modifications to Table 1, adding the mean and SD before and after intervention.

13. Figure 2: Does "current" on the horizontal axis indicate the time of intervention? How is it taken into account in this analysis if there is only one time point before the intervention or one time point after the intervention?

Response: 

Thank you for your comment. We have made mod

---

## [Decision Letter · Decision Letter 1]

31 Jan 2024

PONE-D-23-25847R1The impact of long-term care insurance on family care for older adults: quasi-experimental evidence from ChinaPLOS ONE

Dear Dr. wang,

Thank you for submitting your manuscript to PLOS ONE. After careful consideration, we feel that it has merit but does not fully meet PLOS ONE’s publication criteria as it currently stands. Therefore, we invite you to submit a revised version of the manuscript that addresses the points raised during the review process.

We look forward to receiving your revised manuscript.

Kind regards,

Yuxia Wang

Academic Editor

PLOS ONE

Journal Requirements:

Reviewers' comments:

Reviewer's Responses to Questions

**Comments to the Author**

1. If the authors have adequately addressed your comments raised in a previous round of review and you feel that this manuscript is now acceptable for publication, you may indicate that here to bypass the “Comments to the Author” section, enter your conflict of interest statement in the “Confidential to Editor” section, and submit your "Accept" recommendation.

Reviewer #1: (No Response)

2. Is the manuscript technically sound, and do the data support the conclusions?

Reviewer #1: Yes

3. Has the statistical analysis been performed appropriately and rigorously? 

Reviewer #1: Yes

4. Have the authors made all data underlying the findings in their manuscript fully available?

Reviewer #1: Yes

5. Is the manuscript presented in an intelligible fashion and written in standard English?

Reviewer #1: Yes

6. Review Comments to the Author

Reviewer #1: I would like to express my gratitude to the authors for sincerely addressing the comments and making the necessary amendments. These changes have improved the quality of the manuscript. However, it seems that the description in the Methods section remains insufficient. According to the STROBE guideline, the methods for all sensitivity analyses must also be included in the Methods section. Please consider adding details about Robustness tests, Heterogeneity analysis, and Further analysis in the Methods section. It is better to refrain from presenting additional analytical methods in the results section.

Another concern is the lack of published response rates for the CHARLS data. This omission could potentially introduce a significant bias in the results, so I recommend acknowledging this as a limitation.

Overall, nice work. I hope this research contributes to the development of China's long-term care insurance system.

7. PLOS authors have the option to publish the peer review history of their article (what does this mean?). If published, this will include your full peer review and any attached files.

Reviewer #1: No

---

## [Author Response · Author response to Decision Letter 1]

3 Feb 2024

Response to Minor Revision

Dear Editor and Reviewers,

We would like to express our gratitude for your professional suggestions and constructive comments, which are essential for improving the quality of our research paper. We also greatly appreciate your recognition and acceptance of our paper. Based on some suggestions for minor revisions, we have made the following modifications to our paper:

Reviewer #1: 

I would like to express my gratitude to the authors for sincerely addressing the comments and making the necessary amendments. These changes have improved the quality of the manuscript. However, it seems that the description in the Methods section remains insufficient. According to the STROBE guideline, the methods for all sensitivity analyses must also be included in the Methods section. Please consider adding details about Robustness tests, Heterogeneity analysis, and Further analysis in the Methods section. It is better to refrain from presenting additional analytical methods in the results section.

Another concern is the lack of published response rates for the CHARLS data. This omission could potentially introduce a significant bias in the results, so I recommend acknowledging this as a limitation.

Overall, nice work. I hope this research contributes to the development of China's long-term care insurance system.

Response：

Thank you for your professional feedback and recognition of our manuscript. Your meticulous and professional attitude is commendable. Taking your suggestions into account, we have made the following modifications to the paper:

Firstly, we have reorganized the methodology section based on your suggestions. We have rearranged and elaborated on the methods for different stages in the order of empirical analysis in our paper. We have removed statements describing the methodology in the "Results" section to align it as closely as possible with your requirements.

Secondly, we have added a research limitation regarding the disclosure of response rates in the CHARLS dataset. In fact, based on the data processing results, the response rates for various questions in the CHARLS dataset exceed 90%, ensuring the reliability of the findings in this study.

Other modifications:

We made some slight revisions to the title to ensure a more comprehensive expression of the research content. Additionally, we have also made some formatting adjustments based on previously published papers in this journal.

Finally, we would like to express our heartfelt gratitude for your thorough work. We wish you good health and a happy life in the new year.

---

## [Decision Letter · Decision Letter 2]

20 Feb 2024

The impact of long-term care insurance on family care for older adults: The mediating role of intergenerational financial support

PONE-D-23-25847R2

Dear Dr. wang,

We’re pleased to inform you that your manuscript has been judged scientifically suitable for publication and will be formally accepted for publication once it meets all outstanding technical requirements.

Kind regards,

Yuxia Wang

Academic Editor

PLOS ONE

Additional Editor Comments (optional):

Reviewers' comments:

Reviewer's Responses to Questions

**Comments to the Author**

1. If the authors have adequately addressed your comments raised in a previous round of review and you feel that this manuscript is now acceptable for publication, you may indicate that here to bypass the “Comments to the Author” section, enter your conflict of interest statement in the “Confidential to Editor” section, and submit your "Accept" recommendation.

Reviewer #1: All comments have been addressed

2. Is the manuscript technically sound, and do the data support the conclusions?

Reviewer #1: Yes

3. Has the statistical analysis been performed appropriately and rigorously? 

Reviewer #1: Yes

4. Have the authors made all data underlying the findings in their manuscript fully available?

Reviewer #1: Yes

5. Is the manuscript presented in an intelligible fashion and written in standard English?

Reviewer #1: Yes

6. Review Comments to the Author

Reviewer #1: Thank you for your revisions. No further comments. Great work. I hope this contributes to the development of long-term care insurance system in China.

7. PLOS authors have the option to publish the peer review history of their article (what does this mean?). If published, this will include your full peer review and any attached files.

Reviewer #1: No

---

## [Editor Report · Acceptance letter]

14 May 2024

PONE-D-23-25847R2 

PLOS ONE

Dear Dr. Wang, 

I'm pleased to inform you that your manuscript has been deemed suitable for publication in PLOS ONE. Congratulations! Your manuscript is now being handed over to our production team.

Kind regards, 

on behalf of

Dr. Yuxia Wang 

Academic Editor

PLOS ONE